# Serum Metabolomic and Lipidomic Profiling Reveals Novel Biomarkers of Efficacy for Benfotiamine in Alzheimer’s Disease

**DOI:** 10.3390/ijms222413188

**Published:** 2021-12-07

**Authors:** Ruchika Bhawal, Qin Fu, Elizabeth T. Anderson, Gary E. Gibson, Sheng Zhang

**Affiliations:** 1Proteomics and Metabolomics Facility, Institute of Biotechnology, Cornell University, Ithaca, NY 14853, USA; rb822@cornell.edu (R.B.); qf44@cornell.edu (Q.F.); eta23@cornell.edu (E.T.A.); 2Feil Family Brain and Mind Research Institute, Weill Cornell Medicine, New York, NY 10065, USA; ggibson@med.cornell.edu; 3Burke Neurological Institute, White Plains, NY 10605, USA

**Keywords:** metabolomics, lipidomics, Alzheimer’s disease, serum, thiamine, benfotiamine, mass spectrometry, biomarkers

## Abstract

Serum metabolomics and lipidomics are powerful approaches for discovering unique biomarkers in various diseases and associated therapeutics and for revealing metabolic mechanisms of both. Treatment with Benfotiamine (BFT), a thiamine prodrug, for one year produced encouraging results for patients with mild cognitive impairment and mild Alzheimer’s disease (AD). In this study, a parallel metabolomics and lipidomics approach was applied for the first exploratory investigation on the serum metabolome and lipidome of patients treated with BFT. A total of 315 unique metabolites and 417 lipids species were confidently identified and relatively quantified. Rigorous statistical analyses revealed significant differences between the placebo and BFT treatment groups in 25 metabolites, including thiamine, tyrosine, tryptophan, lysine, and 22 lipid species, mostly belonging to phosphatidylcholines. Additionally, 10 of 11 metabolites and 14 of 15 lipid species reported in previous literature to follow AD progression changed in the opposite direction to those reported to reflect AD progression. Enrichment and pathway analyses show that significantly altered metabolites by BFT are involved in glucose metabolism and biosynthesis of aromatic amino acids. Our study discovered that multiple novel biomarkers and multiple mechanisms that may underlie the benefit of BFT are potential therapeutic targets in AD and should be validated in studies with larger sample sizes.

## 1. Introduction

Thiamine (vitamin B1) is essential for brain function because of its role as the coenzyme thiamine diphosphate (ThDP) in glucose and energy metabolism and multiple non-coenzyme -functions [1]. Evidence in animals and humans linking abnormalities in thiamine availability and metabolism to the pathophysiology of AD has brought attention to the regulation of thiamine as a therapeutic target for treating AD. Thiamine is a water-soluble vitamin unable to cross biological membranes in the absence of a transport protein [2,3]. Therefore, compounds such as benfotiamine (BFT), a thiamine prodrug, were developed to increase the thiamine to levels not possible with thiamine itself [4]. In animal models, thiamine deficiency promotes AD pathology, while BFT diminishes pathology [5]. This insight motivated us to complete a recent pilot clinical trial of BFT in AD patients. BFT produced encouraging results for effective treatment of patients with mild cognitive impairment and/or mild AD [6]. BFT is currently in clinical trials for large-scale AD patients’ therapy.

Despite the encouraging results using BFT for the treatment for patients with mild cognitive deficits or mild AD [6,7,8], important gaps exist in our mechanistic understanding of BFT treatment that need to be addressed to improve clinical outcome and development of new drugs for AD. AD is defined by abnormal extracellular accumulation of amyloid-*β* peptide (A*β*) in amyloid plaques, tau protein aggregated in intracellular neurofibrillary tangles (NFTs), and neurodegeneration, which is inferred from decreased glucose metabolism measured by FDGPET [9,10]. However, many other changes occur, including inflammation, oxidative stress, brain atrophy, and vascular changes [11,12,13]. The lack of any effective drugs being developed against the disease along with up to 400 failed therapeutic trials indicates the lack of mechanistic understanding of AD pathogenesis and its progression.

Mass spectrometry (MS)-based metabolomics technologies hold promise for discovery of therapeutic targets and biomarkers for disease diagnosis, prognosis, and monitoring the effects of therapeutics [14]. However, identification of the complete metabolome in any biological system remains difficult due to the lack of appropriate public metabolome databases containing both a high quality of fragmentation spectra and accurate mass and retention time (AMRT) for all metabolites, allowing for an unambiguous and reliable identity [15]. The other limitation is that many metabolites are detected in multiple forms (e.g., adducts with different ions of salts and solvents) leading to the formation of many derivatives with an enormous complexity and difficulty of annotation. As a result, the overwhelming majority of MS-detected features remain unidentified, and more complete identification of compounds requires significant effort. This has led to the emergence of new comprehensive approaches to improve the identification of compounds. Among them, the annotation (i.e., low probability identification) of compounds based on their involvement in a particular biological pathway for various diseases such as AD have been developed [16,17,18] and an in-house reliable library for combination of MS2 spectra and AMRT has been created and used [15]. Lipidomics, a subset of the metabolomics field, is another liquid chromatography (LC)–MS-based analytical approach that focuses on identifying and profiling the full complement of lipid classes, subclasses, and lipid species in complex biological samples. An integration of metabolomics with lipidomics provides global overview of changes in metabolites and lipids and their associated metabolic pathways [19,20]. The proximity of lipidomics and metabolomics to the biological phenotypes integrates the cascading effects of the environment, gene expression, and regulatory processes. Traditional biochemical and pharmacological approaches for determining mechanism fail when applied to slowly progressing diseases such as AD or long-term treatments, but metabolomics/lipidomics show promise, because metabolites reflect ongoing changes in chronic conditions [18,21].

Changes in metabolism and lipids levels in serum and/or brain have been implicated in AD [22,23,24,25]. However, none of these have been explored in AD serum after a 12-month treatment with BFT. Thus, we are particularly interested in using this exploratory study to screen specific BFT-responsive metabolites and lipids and test if some of those metabolites/lipids that were reported previously in AD progression are changed by BFT treatment. The reversal of change in level of these metabolites and lipids may be excellent indicators or biomarkers for evaluation of effective BFT treatment.

## 2. Results

### 2.1. Study Characteristics and Parallel Metabolomics and Lipidomics Workflows

One of the primary objectives was to use this exploratory study to test our liquid chromatographic MS workflows for parallel serum metabolomic and lipidomic analyses using our in-house built compound spectral libraries to maximize the number of metabolites/lipids identification with enhanced reliability and confidence [24] and to test our ability to make reproducible measures in small volumes of serum. Figure 1 shows a schematic diagram of the experimental design and analytical workflows implemented in this study. Serum samples from ten patients with mild cognitive impairment or mild AD that were treated with placebo (*n* = 5) or BFT (*n* = 5) were analyzed (see details in the Materials and Methods section).

The untargeted metabolomics analysis revealed 8713 and 3710 metabolic features by C18 and hydrophilic interaction liquid chromatography (HILIC) LC–MS workflows, respectively. Upon annotation by a database processing search in Compound Discoverer (CD) (see Methods for details) and subsequent filtering by removing background compounds, reducing redundancy in annotation, and normalizing QCs area values across all samples, 284 and 103 metabolites were confidently identified and annotated in C18 and HILIC analyses, respectively. All the identified unique metabolites have high quality MS2 spectra for preferred ions with the best batch score > 60 against either public mzCloud database or our in-house spectral mzVault libraries. By combining C18 and HILIC results and removing the duplicate compounds, we identified 315 unique metabolites. These 315 metabolites represented diverse metabolite classes such as amino acids, short peptides and analogues, lipids and lipid-like molecules (including fatty acyls and steroids), as well as carboxylic acids and derivatives. A summary table for all unique metabolites identified in untargeted metabolomics analysis after filtering and used for further statistical treatment with various levels of confidence is shown in Appendix A.

Untargeted lipidomics analysis identified 292 lipid forms in five major lipid classes, phosphatidylcholines (PC), lysophosphatidylcholines (LPC), sphingomyelins (SM), diacylglycerols (DG), and triacylglycerols (TG), in positive ion mode by LipidSearch software. We also identified 140 lipid forms by LC–MS/MS under negative ion mode, which come from six lipid classes: phosphatidylcholines (PC), lysophosphatidylcholines (LPC), sphingomyelins (SM), phosphatidylethanolamines (PE) phosphatidylinositol (PI), and ceramides (Cer). Combining lipid species from both positive and negative ion modes and removing the duplicates, we identified a total of 417 unique lipid species. A summary table for all lipid species identified in untargeted lipidomics after filtering and used for further statistical analysis is shown in Appendix A.

### 2.2. Metabolic Profiling Distinguishes the BFT Treatment from Placebo Groups

We applied the multivariate statistical analysis for the 284 and 103 annotated metabolites from C18 and HILIC, respectively, after Pareto scaling (Par) and log transformation. After transformation of the data, they were used for supervised OPLS-DA (orthogonal partial least square differential analysis) model analysis. Next, we created a score plot to visualize the OPLS-DA model and characterize the contribution of variables to the separation of classes using the loading plot that provided VIP scores, and the OPLS-DA score plots reveal remarkable differences among QC, placebo and BFT sample groups [26] (Figure 2). As expected, QC samples containing equal aliquots of all the treated and placebo samples also form a tight and distinct cluster. The results in Figure 2 demonstrate that our LC–MS analytical workflows and methods yielded high reproducibility based on QC samples and with a confident data quality despite the limited five biological replicates available in each group in this exploratory study. The OPLS-DA models indicate a clear separation of BFT treatment (red dots) from placebo (green dots) groups and QC samples that were represented by blue dots in both Figure 2A (R^2^X = 0.415, R^2^Y = 0.978, Q^2^ = 0.351) and Figure 2B (R^2^X = 0.617, R^2^Y = 0.914, Q^2^ = 0.344). Q^2^ and R^2^ indicate the robustness of the model. 

An OPLS-DA S-plot used to identify the biologically relevant changes in metabolomic dataset are shown in Appendix A, and the potential markers were extracted based on their contribution to the variations and correlation within the dataset. Based on the criterion of VIP ≥ 1, the most relevant metabolites for contribution to the differences [27], 25 metabolites were significantly different between the placebo and BFT treatment groups. A summary of identified metabolites with largest fold changes between the two groups are listed in Table 1. Out of the altered 25 metabolites, 10 were significantly increased in the BFT treatment group, including thiamine, indole, tyrosine, and tryptophan, while 15 were decreased, including O-Ureido-D-serine, lysine, and androsterone sulfate. A final column (literature FC for AD/C) was added for comparison of our results to previously reported changes of the metabolites in AD progression. The arrows pointing in the opposite direction indicate that BFT reversed changes associated with AD progression.

Cluster heat maps were generated for the 284 metabolites identified in C18 mode and 103 in HILIC mode for data visualization of identified metabolites with statistical changes between placebo and the BFT treated samples. Figure 3A,B represent the hierarchical clustering maps for the top 25 metabolites with significant changes in relative abundance with *p* < 0.1 using *t*-test/ANOVA from C18 and HILIC analyses, respectively. The 25 top metabolites identified in heat map overlaps mostly with the VIP score in Table 1. This result shows that our data are consistent in different statistical treatments.

### 2.3. Significant Changes of Phosphatidylcholine, Sphingomyelin and Triglycerides Species Levels in BFT Treated Group

The impact of BFT on the serum lipid profile was also determined. A total of 417 lipids in 7 major categories were confidently identified by LC–MS/MS and, subsequently, LipidSearch 4.2 software with an integrated database of Lipid Search. A summary table for all lipids of major lipid classes that were identified with various levels of confidence is shown in Appendix A. A supervised OPLS-DA plot for all identified 417 lipids and a heatmap for 50 lipids with largest change in serum were shown in Figure 4A,B, respectively, after combining results from both positive and negative ion modes of the LC–MS/MS analyses. The OPLS-DA models indicate a clear separation between treatment (green dots) and placebo (red dots) groups in Figure 4A (R^2^X = 0.832, R^2^Y = 0.998, Q^2^ = 0.681), while the heat map shows top 50 lipids that differ significantly in relative abundance with *p* < 0.1 using *t*-test/ANOVA between FBT treatment and placebo serum samples (Figure 4B).

Based on the criterion of VIP ≥ 1, 22 lipid forms with largest fold changes and significantly different between the placebo and BFT treatment groups are listed in Table 2. Of the altered 22 lipids, 16 were significantly increased in the BFT treatment group mainly belonging to PC and SM categories, while 6 were decreased in TGs. Phosphatidylcholines lipid species change dramatically with BFT treatment, and this was not predictable. Fourteen of the lipids changed in the opposite direction reported in the literature, suggesting BFT was reversing AD-related changes. Only one lipid with fold changes in the same direction was found in AD progression. Further analysis showed that increased phosphatidylcholines and sphingomyelins along with decreased triglycerides were the main categories of lipids yielding clear differences in the serum lipid profiles in response to BFT treatment, and no significant changes in the level of other lipid classes were observed, as shown in Figure 5.

### 2.4. Identification of Metabolic, Lipid Signatures and Biomarker Candidates

Univariate ROC analysis was conducted to further characterize the predictive value of these individual metabolites independently. Of particular interest are four metabolites (thiamine, L-Tryptophan, Indole, and L-tyrosine with an area under the curve (AUC) > 0.7 that was significantly increased in treated serum samples, including the most significant one, thiamine, as expected, with a 100-fold increase (Figure 6A and Table 1). Thiamine has a log_2_(fold change) value of 6.66, with *p* value < 0.002, while both L-tryptophan and Indole show 1.6-fold increase with *p* value < 0.03. L-tyrosine increases by 1.7-fold with *p* value < 0.05. We also found that several metabolites of particular interest (e.g., Lysine, O-ureido-D-serine, androsterone sulfate, and 16 hydroxy decanoic acid) were significantly decreased in treated samples compared to placebo samples, 0.08-, 0.75-, 0.73-, and 0.48-fold, respectively (Figure 6B).

We found six lipids for which PC(38:5), PC(32:2), and PC(36:3) were increased in treated serum samples by 2.6-fold, *p* value < 0.005, 3.7-fold, *p* value < 0.001, and 3.6-fold, *p* value < 0.01, respectively; three of them belonged to sphingomyelins–SM(d40:2), SM(D41:1), and SM(d39:2) with increase by 1.3-fold, *p* value < 0.05, 1.5-fold, *p* value < 0.02, and 1.7-fold, *p* value < 0.05, respectively (Figure 7). Most strikingly, phosphatidylcholine species levels are increased significantly in serum as opposed to their levels declining in AD progression, as found in earlier studies (see Table 2). We found that levels of several TG lipids decrease consistently by ~1.5-fold after benfotiamine treatment (Table 2 and Appendix A).

### 2.5. Pathway and Enrichment Analysis of Altered Metabolites

Metabolite set enrichment analysis (MSEA) is a method designed to identify and interpret patterns of metabolite concentration changes in a biologically meaningful way. Pathway analysis helps to understand which biological pathways, representing collections of molecules performing a particular function, may be involved in response to a disease phenotype or drug treatment. Enrichment analysis for metabolite datasets based on KEGG human metabolic pathways revealed major changes in few specific pathways such as thiamine metabolism and the biosynthesis of phenylalanine, tyrosine, tryptophan, and glutathione metabolism after a 12-month trial period of BFT for MCI/mild AD patients compared to placebo (Appendix A). All these metabolite sets have at least four entries found. Furthermore, pathway analysis revealed that these potential biomarkers are mainly involved in thiamine metabolism, phenylalanine, tyrosine, tryptophan biosynthesis, glutathione metabolism, and linoleic acid metabolism (Appendix A). In summary, these involved pathways provide more insightful understandings of the potential metabolic processes associated with BFT treatment (Appendix A).

## 3. Discussion

Metabolomics is one of the latest “omics” lexicons and a rapidly emerging technology in the field of life sciences including biomedical research and precision medicine [14]. With current advances in mass spectrometry technologies, metabolomics studies enable the identification of novel metabolites and provide a more detailed characterization of biological pathways in many organisms. Application of metabolomics and lipidomics in biomedical research has leveraged many biomarkers’ discovery in various disease such as AD [12,34] and cancers [24] and their therapeutics [35,36]. Specifically, global metabolomics and lipidomics have widely adopted LC–MS for analysis of serum samples to identify the differences in metabolites levels in various diseases during their progression or during the treatment with drugs [37]. Previous studies demonstrated that specific altered metabolites and lipids in plasma samples varied with AD progression [18,34,38]. Confident identification of low abundant metabolites or biomarkers is still one of the bottlenecks particularly in serum samples due to the high complexity and extraordinarily wide dynamic range of analytes that interfere with identification by LC–MS analysis. In this work, we found and identified multiple exogenous drugs and their metabolic products that appeared responsive to BFT, which is not surprising since AD patients are older and tend to take multiple drugs. This suggests that BFT alters drug metabolism, but that study is beyond the scope of this manuscript. Another common limitation of current untargeted metabolomics technology is the relatively high false discovery rate for accurate annotation of those identified metabolites due to the lack of additional retention time information in most public databases. It should be noted that pre-analytical sample processing is of importance to the metabolic integrity of serum samples used in this study. We strictly followed the best practices for sample storage, handling, and shipping prior to analysis (see details in Method section).

In this study, we took advantage of our in-house constructed spectral libraries covering 642 and 479 standard compounds for C18 and HILIC workflows, respectively, which were used for database processing along with mzCloud database. We found that approximately 15% of 315 unique metabolites were identified only in our in-house spectral libraries (no match to any public databases). In combination with high cutoff scores for filtering out initially identified metabolites, we found the 315 metabolites reported in this study are identified with a high degree of confidence.

Several studies found that thiamine levels and the activity of thiamine-dependent enzymes are reduced in the brains and peripheral tissues of patients with AD [8,39]. Therefore, increasing the levels of thiamine in blood and brain is essential to preventing the cognitive impairment in AD [6,38]. Another metabolite that showed the increase in serum of BFT treated samples is L-tryptophan. The data are also in agreement with the pathway analysis of the metabolites, where tryptophan, tyrosine, and phenylalanine biosynthesis pathway show a major impact (see Appendix A). In our study, we found that levels of tryptophan and indole were increased by 1.6-fold after BFT administration for 12 months, while both metabolites were decreased in AD progression (Table 1). Tryptophan is the amino acid needed to synthesize proteins and neurotransmitters such as serotonin, niacin, and melatonin. Increasing serum levels of tryptophan reflect either an increase in tryptophan biosynthesis pathway or a decrease in tryptophan metabolism [40]. It was found that serotonergic signaling disruption and systemic inflammation have been associated with the pathogenesis of AD, and tryptophan metabolism is the responsible factor [41]. Low levels of L-tryptophan reflect cognitive performance and AD, suggesting the bioavailability of the neurotransmitter decreased [42,43]. Our data show that serum tryptophan levels increase, potentially due to activation of the biosynthesis pathway by BFT. In the cytosol, as we know, thiamine acts as a cofactor for TKT, a key enzyme of the pentose phosphate pathway (PPP). The PPP provides precursors for nucleotide and amino acid biosynthesis for tryptophan, phenylalanine, and tyrosine. The same increased effect by BFT was also observed for tyrosine and histidine, as shown in Table 1. L-tyrosine shows an increase by 1.7-fold with BFT treatment in patient. Although there are only few studies measuring tyrosine levels specifically in AD brain and serum, several studies did report that oral tyrosine administration improves memory and cognitive function [44,45] in mild AD patients [46]. The higher levels of tyrosine and tryptophan in serum suggest that reversal of these levels is one mechanism by which BFT is beneficial. Choline is another metabolite that shows a fold change of 1.77 with BFT treatment; even its *p* value (0.06) is not that significant, which might be one of the factors for the increase in phosphatidyl choline levels in serum. We acknowledge that this is an exploratory pilot study, and thus, a larger study on more samples will be warranted to test the validity of our results.

On the other hand, we found that lysine levels decrease dramatically by more than 10-fold with the intake of BFT for a prolonged time, while in AD, it was found to increase in plasma levels (Table 1) [29,47]. It was also found that lysine level in serum increases with thiamine deficiency, which, in turn, increases stress-induced anxiety level [48]. Oxoacid dehydrogenase (OADHc) is involved in lysine-derived 2-oxoadipate catabolism [49], which also required thiamine diphosphate as a cofactor to form glutaryl CoA. The data consistently showed that thiamine-dependent enzyme activity changes with BFT treatment and results in either an increase or a decrease in metabolites in those pathways. O-Ureido-D-Serine level was found to be decreased by 1.33-fold in serum sample of BFT treated individuals. This is a derivative of D-serine, a signaling molecule involved in neurotransmission. However, it was observed in previous studies that serum D-Serine level significantly increased with the early progression of AD [50,51]. The direct mechanism of benfotiamine that leads to the decreased D-serine is not clear. 16-hydroxyhexadeconoic acid is a hydroxy long chain fatty acid that decreases significantly in serum with BFT, suggesting that it might be due to alpha oxidation of hydroxy long chain fatty acid, as studies have found that α oxidation of fatty acids in peroxisomes by 2-hydroxyacyl-CoA lyase (HCAL1) enzyme uses thiamine pyrophosphate (TPP) [52].

Our biomarker discovery results show that, out of 25 metabolites with significant changes in this study, 11 of them were also reported to be associated with AD progression, as shown in Table 1. Strikingly, 10 out of 11 of those metabolites show inverse changes of BFT treatment when compared to those found in AD progression, suggesting that these 10 could be used as candidate biomarkers for monitoring the effectiveness of BFT treatment in AD after further validation with large sample size.

We extended our studies on the lipidomics profiling of the same samples and found that BFT increased three major sphingomyelins with *p* value less than 0.05-SM(d40:2) SM(d41:1) and SM (d39:2) and many phosphatidylcholine lipids species (PC(38:5) PC(32:2), PC(32:1), PC(40:6), PC(40:4), PC(34:4), PC(36:5), and PC(36:3)), as shown in Table 2. We anticipated that phosphatidyl level increases in serum (Table 2) may be either due to increases in choline levels (Table 1) or may be due to the inhibition of arachidonic acid pathway by BFT, which blocks the NADPH oxidase (NOX) and cyclooxygenase (COX) enzymes [53]. Lipids are tightly connected with metabolism of the amyloid precursor protein (APP), which produces amyloid-beta peptide (Aβ), the main component of senile plaques, but the mechanism is not clear. Significant differences in plasma lipids level of some phosphocholines, sphingomyelins, and triglycerides between AD and control were observed in both blood and brain [47,54]. Many prior studies measure the levels of phosphatidylcholine in plasma and reported that significant decrease in accelerated aging and AD [33,55], while we found remarkable increases in the overwhelming majority of phosphatidylcholine species with BFT treatment. Additionally, 15 of the top 22 lipid forms with largest change in this study were previously reported to be associated with AD progression. Again, 14 out of 15 those lipid species show reverse changes of BFT treatment versus those found in AD progression, suggesting these 14 lipid forms could be used as candidate biomarkers for assessing the effect of BFT treatment in AD after additional validation with large sample size.

Taking together all the data, here, we propose that several metabolic pathways are associated with the treatment of benfotiamine for mild AD patients after a prolonged period of 12 months. The dramatic increase in the thiamine levels in serum with benfotiamine can alter multiple pathways (Figure 8).

It should be emphasized that this exploratory study was conducted with a small sample size available from AD patients with and without BFT treatment. Therefore, the reported findings are observational and need further design of validation experiments in a larger sample size and in combination with other omics technologies as well as biochemical analyses for drawing direct conclusions. Even the results due to either associated other diseases or changes in synthesis or degradation of other drugs or other metabolites related to intake of dietary supplements will be informative if replicable. Despite the above limitations, our serum metabolomic and lipidomic datasets on the control placebo group and BFT-treated group show significant and large differences in amino acids, fatty acids, sphingomyelins, phosphatidylcholines, and several metabolites, and our statistical analysis demonstrates that the data are reliable and convincing with relatively high quality. It is particularly important to point out that the potential biomarkers with most significant changes in BFT treatment and their associated pathways were found to be consistently in the direction of benefit for BFT treatment in mild AD progression.

## 4. Materials and Methods

### 4.1. Serum Sample Collection for Metabolomics and Lipidomics

Mild dementia patients due to AD were enrolled to the Burke Rehabilitation Hospital outpatient clinic [6]. The trial duration per participant was twelve months. Participants in the treated group took one 300 mg capsule of Benfotiamine in the morning and one in the evening. The participants in the placebo group took one 300 mg capsule in the morning and evening with microcrystalline cellulose without BFT. The BFT and placebo were manufactured and provided by the Advanced Orthomolecular Research, Canada. They prepared the BFT according to an FDA-approved IND, which was prepared by the Cornell Translational Science Center and issued to the Burke Neurological Institute. The trial was approved by the Institutional Review Boards of the Burke Rehabilitation Hospital and Weill Cornell Medicine, in which aside samples for exploratory measures were approved by the IRBs with the protocol number: BRC-451. In this study, 10 serum samples collected from 5 of treatment and 5 of placebo group after 12 months of BFT trial were available. We followed the recommendation and best practices for pre-analytical processing of the serum samples for metabolomics analysis [56]. Serum samples were stored at −80 °C for 24–60 months in Burke Neurological Institute and shipped in a dry ice package to the Cornell Proteomics and Metabolomics Facility for metabolomics/lipidomics analysis. Samples were stored at −80 °C in the facility for 3 weeks and thawed for untargeted metabolomic studies (100 µL of serum) and untargeted lipidomics analysis (30 µL of serum).

### 4.2. Untargeted Metabolomics Analysis of Serum Samples

Briefly, we added cold methanol (300 µL) to 100 µL of each serum sample to precipitate protein at 4 °C; three QC samples were prepared with one global QC by mixing equal aliquot of 10 serum samples, and two group QC samples were prepared by mixing equal aliquot of the 5 samples from each group. After protein precipitation and collection of supernatants, each supernatant sample was divided into two equal halves and then dried down for C18-based non-polar metabolites and HILIC-based polar metabolites identification. Each dried sample was reconstituted in 80 μL of 20% ACN, 0.1% formic acid solvent containing ^13^C valine, ^13^Pyruvic acid, and Sulfadimethoxine as internal standards for monitoring the instrument performance overtime. HILIC chromatographic separation was performed on a Vanquish UHPLC system with a SeQuant ZIC pHILIC column (5 µm, 2.1 × 150 mm) coupled to a Q Exactive™ Hybrid Quadrupole-Orbitrap High Resolution Mass Spectrometer (Thermo Fisher Scientific, San Jose, CA, USA). The mobile phase consisted of (A) 10 mM ammonium acetate in water pH = 9.8 with 0.1% formic acid and (B) acetonitrile. The gradient was as follows: 0–15 min, 90–30% solvent B; 15–18 min, isocratic 30% solvent B; 18–19 min, 30–90% solvent B; 19–27 min, 90% solvent B; followed by 3 min of equilibration of the column before the next run. The flow rate was 250 μL/min, and the injection volumes were set to 2 μL. C18 chromatographic separation was performed on the same Vanquish UHPLC-QE-HF system with an Accucore Vanquish C18+, 1.5 μm column (2.1 mm id × 100 mm). The detailed method can be found in the published literature [57]. The acquired datasets for C18 and HILIC mode, composed of full MS and data-dependent MS–MS raw files under both positive and negative ion modes, were processed using CD 3.2 using the workflow mentioned in the literature [57]. The peak areas and retention time for each of internal standards spiked in each sample allowed us to assess sample preparation, samples’ matrix effect, and LC–MS technical variations.

### 4.3. Metabolomics Data Processing and Statistical Analysis

Normalization of all LC–MS raw data was based on total signal normalization step against global QC files by Compound Discoverer (CD) 3.2 software (Thermo Fisher Scientific, San Jose, CA, USA), which calculates the total intensity of the whole spectrum or total ion count and sets it to a constant value. Group QC samples were used for identification of metabolites based on MS and MS2 spectra against the public databases as well as additional retention time against our in-house spectral libraries. Two in-house-built spectral libraries (642 compounds for C18 and 479 for HILIC) having AMRT with high quality of MS2 spectra along with public mzCloud database (accessed on 9 July 2021) were used to annotate compounds on MS/MS level with a mass tolerance of 10 ppm. Additional databases including ChemSpider, BioCyc, Human Metabolome Database, and KEGG database (accessed on 9 July 2021) were used to annotate features based on exact mass with a mass tolerance of 5 ppm as well as the CD internal database (an endogenous metabolites database of 4400 compounds). The initially identified molecules in serum samples were filtered out in CD through background subtraction and exclusion of false positive or repetitive features without MS2 spectra and removal of compounds not found in QC samples [58]. All identified unique metabolites have high quality MS2 spectra for preferred ions with the best batch score > 60 against either mzCloud database or our in-house spectral mzVault libraries.

To maximize identification of differences in metabolic profiles between groups in both C18 and HILIC analyses, the orthogonal-projection-to-latent-structure–discriminant-analysis (OPLS-DA) model was applied using the SIMCA-P software (version 13.0, Umetrics AB, Umea, Sweden). The variable importance in the projection (VIP) value of each variable in the model was calculated to indicate its contribution to the classification. A higher VIP value represented a stronger contribution to discrimination among groups. VIP > 1.0 were considered significantly different. After that, we combined both a C18- and HILIC-annotated compound list to perform both univariate analysis for ROC plots and metabolite enrichment pathway analysis. Log transformation of the normalized peak intensity values and pareto scaling were performed before statistical analysis to obtain the enrichment analysis map and identify the potential biomarkers using ROC curve in MetaboAnalyst 5.0 software https://www.metaboanalyst.ca/ (accessed on 9 July 2021).

### 4.4. Untargeted Lipidomics Analysis of Serum Samples

The internal standard mixture (ISTD) spiked into each serum sample before extraction consisted of 25 μg/mL of TG (15:0)_3_, PC (18:1d7-15:0), PG (14:0)_2_, LysoPC (18:1-d7), PS (16:0)_2_, Ceramide (d18:1/12:0), and 17:0 Cholesteryl ester. ISTD mixture (30 μL) was added to each 30 μL serum sample, followed by 190 μL of MeOH. Samples were then processed and analyzed using C30 column, using the same gradient in similar way as mentioned in the literature [20]. Specifically, chromatographic separation was performed on the same Vanquish UHPLC/QE-HF system with an Accucore C30, 2.6 μm column (2.1 mm id × 150 mm). The mobile phase consisted of (solution A) 60% ACN, 40% water, 10 mM ammonium formate with 0.1% formic acid and (solution B) 90% IPA, 10% ACN, 10 mM ammonium formate with 0.1% formic acid. The gradient was as follows: 0–1.5 min, 32% solvent B; 1.5–4 min,32–45% B; 4–5 min, 45–52% B; 5–8 min 52–58% B; 8–11 min, 58–66% B; 11–14 min, 66–70% B; 14–18 min, 70–75% B; 18–21 min, 75–97%s B; 21–25 min, 97% B; 25–25.1 min 97–32% B; followed by 4 min of re-equilibration of the column before the next run. The flow rate was 260 μL/min, and 2 μL of each sample was injected. Acquired MS/MS data were processed using LipidSearch™ software version 4.2 (Thermo Scientific, San Jose, CA, USA) against an online MS/MS lipid library. Data obtained from lipid search software were then generalized, log-transformed, pareto-scaled, and subjected to partial least-squares discriminant analysis and analysis of variance with an adjusted *p* value (false discovery rate) cutoff of 0.05 and post hoc analyses by Fisher’s least significant difference test. Heat maps were generated using Euclidean distance measure and Ward clustering algorithm. All lipidomics data analyses were performed in MetaboAnalyst 5.0 software.

## 5. Conclusions

The untargeted serum metabolomics and lipidomics implemented with advancement of mass spectrometric technologies and use of our in-house compound spectral library demonstrated a powerful approach for leveraging confidence in identification and correct annotation of metabolites in global metabolomics/lipidomics. Our study revealed dozens of metabolites/lipids as novel potential biomarkers that are closely related to the response of benfotiamine treatment for 12 months and could be used for assessment of efficacy for benfotiamine treatment in AD. Responsive changes to BFT treatment for two dozen of biomarker candidates were found to be the complete reversal of the changes observed in the AD progression. All phosphatidylcholine lipids species have shown significant increases in serum level with BFT, while the decrease in those phosphatidylcholines was reported in AD progression. The results suggest that potential mechanistic pathways underlie the benefit of BFT in AD. This is the first report of serum metabolomic and lipidomic investigation of the metabolic changes in serum following intake of an analogue of thiamine for treatment of mild AD or dementia patients. Future validation of the identified biomarker candidates and associated mechanistic pathways of benfotiamine will need to be investigated.

## Figures and Tables

**Figure 1 ijms-22-13188-f001:**
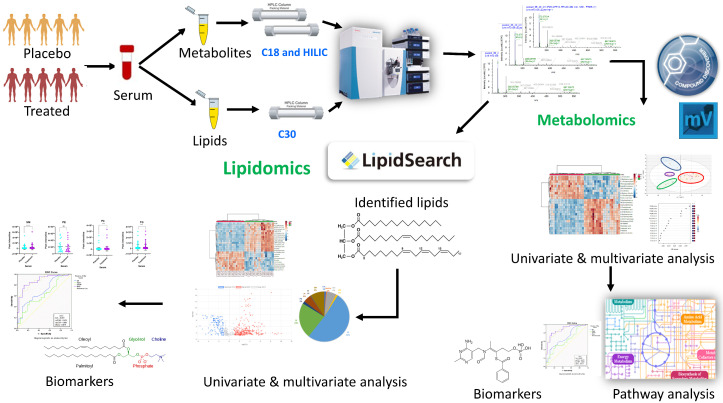
Schematic diagram of the parallel metabolomic and lipidomic workflows used in this study. Serum samples were collected from patients, and aliquoted for separate extraction of metabolites and lipids. The extracted samples were analyzed by LC-MS/MS using C18 and HILIC columns (metabolomics) and C30 column (lipidomics). Raw MS and MS/MS data files were acquired by high resolution mass spectrometer on Orbitrap QE-HF and processed by Compound Discoverer (metabolomics) and LipidSearch (lipidomics). Further statistical and bioinformatics analyses were performed to identify altered metabolites/lipids and potential mechanistic pathways in response to benfotiamine treatment.

**Figure 2 ijms-22-13188-f002:**
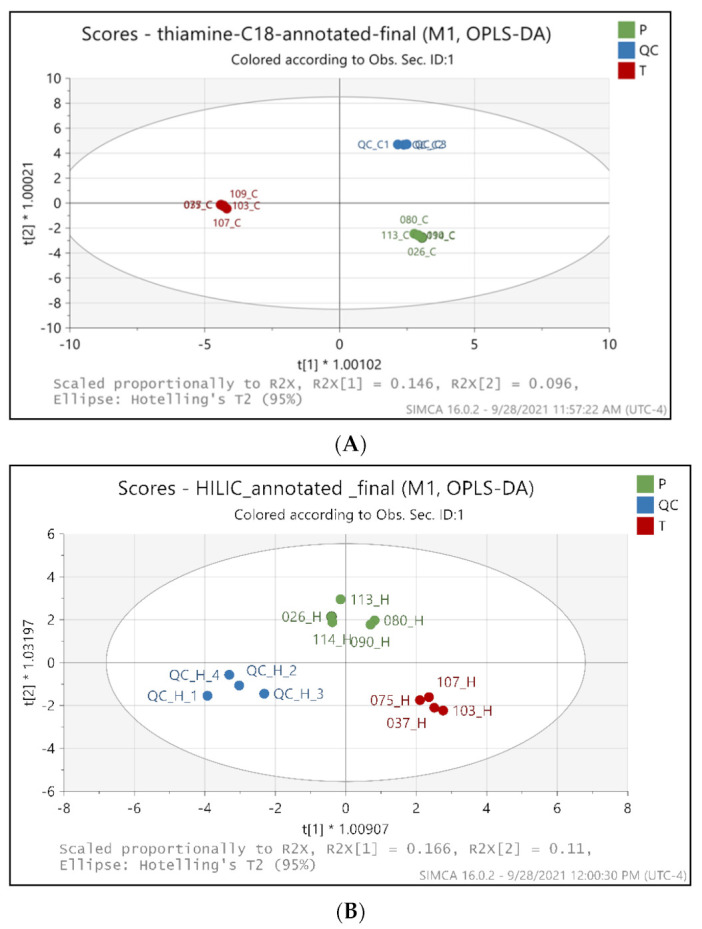
OPLS-DA score plots between the metabolites identified in placebo and benfotiamine treated serum samples in C18 mode (**A**) and HILIC mode (**B**). Green dots-Placebo samples, Red dots-benfotiamine-treated samples, Blue dots-QC pooled samples. Each dot represents a sample. P-Placebo, T-benfotiamine treated.

**Figure 3 ijms-22-13188-f003:**
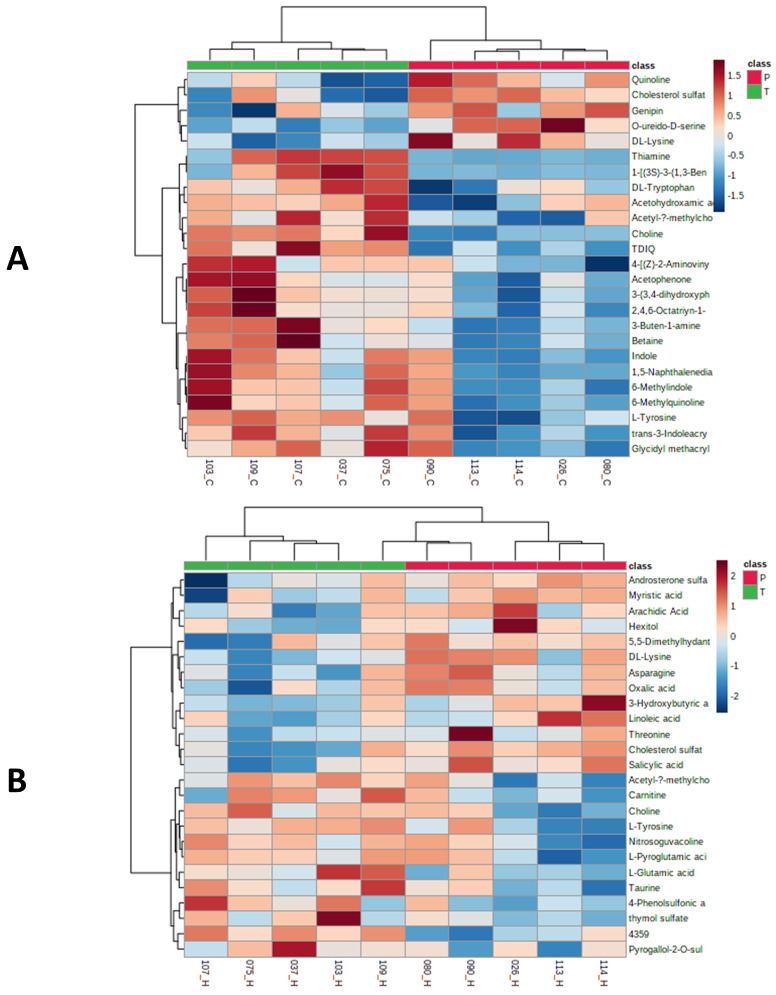
Heat maps showed the distribution of top 25 metabolites that were significantly different between placebo and treated serum samples by C18 (**A**) and HILIC (**B**) analyses. The serum samples from placebo and treatment groups were labeled with red and green ribbons, respectively. The heatmap scale ranges from-2 to 2 in log_10_ (auto scaled intensities) scales (KEGG pathway metabolites).

**Figure 4 ijms-22-13188-f004:**
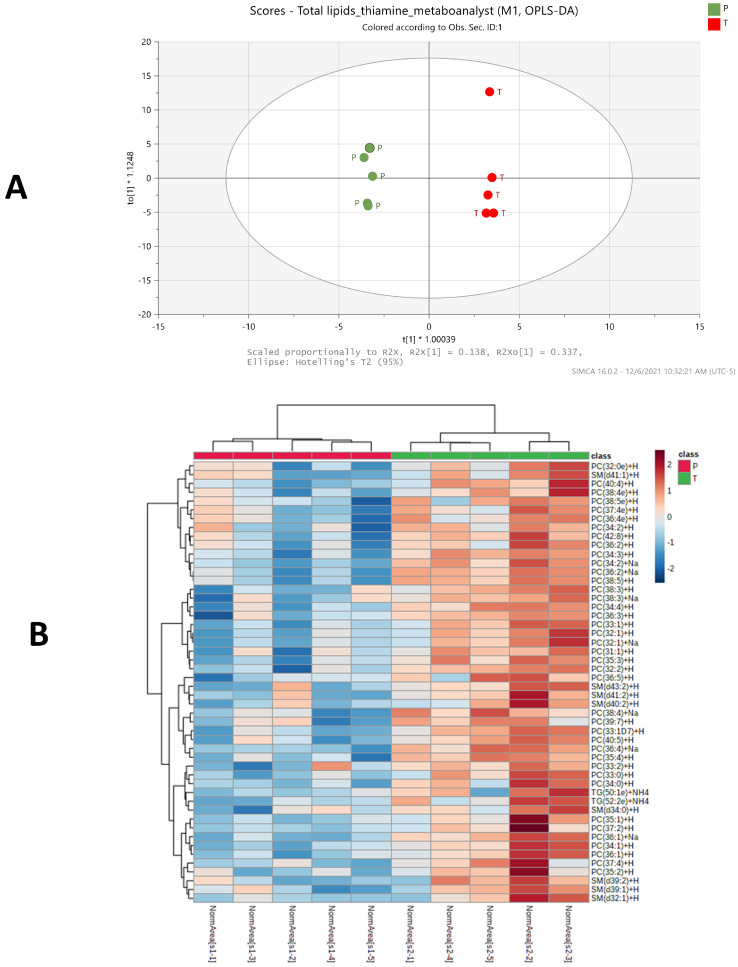
Lipidomic profiling distinguishes the BFT treatment from placebo group in an OPLS-DA score plot between the placebo, treatment serum samples (**A**), and a heatmap showing the distribution of top 50 lipids that were significantly different between placebo and treatment serum samples, PC-Phosphatidylcholine, SM-Sphingomyelins, TG-Triglycerides. (**B**). The serum samples from placebo and treatment groups were labeled with red and green ribbons, respectively. The heatmap scale ranges from −2 to 2 in log_2_ (auto scaled intensities) scales (Lipidomics analysis metabolites).

**Figure 5 ijms-22-13188-f005:**
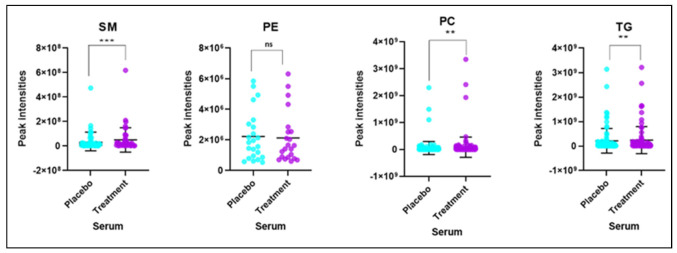
Intensities of different lipid classes or species in serum samples of placebo and benfotiamine treated mild AD (ns = not significant, ** *p* value <0.01, *** *p* value < 0.001), SM-Sphingomyelins, PE-Phosphatidylethanolamine, PC-Phosphatidylcholine, and TG-Triglycerides. Each dot represents a lipid species.

**Figure 6 ijms-22-13188-f006:**
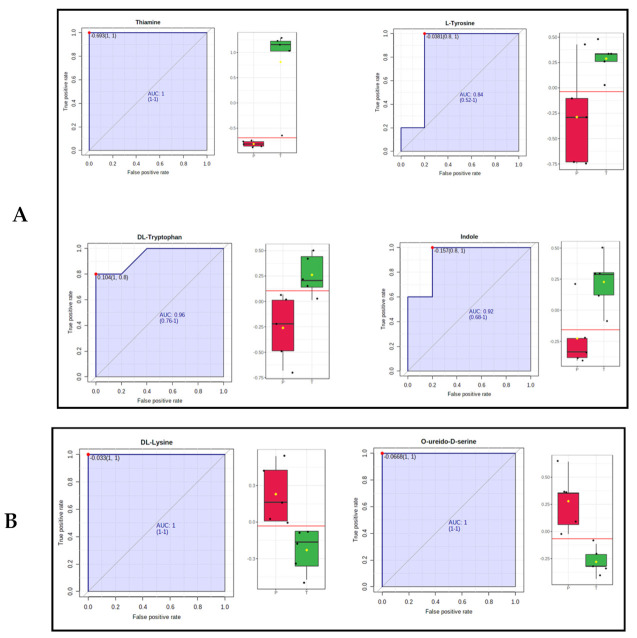
Univariate ROC analysis of the significantly changed metabolites. (**A**) ROC plots for four representative metabolites (Thiamine, L-tyrosine, L-Tryptophan and Indole) that increase in benfotiamine treatment serum samples (*p* value < 0.05). (**B**) ROC plots for three representative metabolites (O ureido-D-Seine, carboxymethyl-lysine) that decrease in benfotiamine treatment serum samples (*p* value < 0.05). Red boxes represent placebo samples, green boxes are BFT treated samples and yellow diamond denotes the median value for log_2_ (fold change).

**Figure 7 ijms-22-13188-f007:**
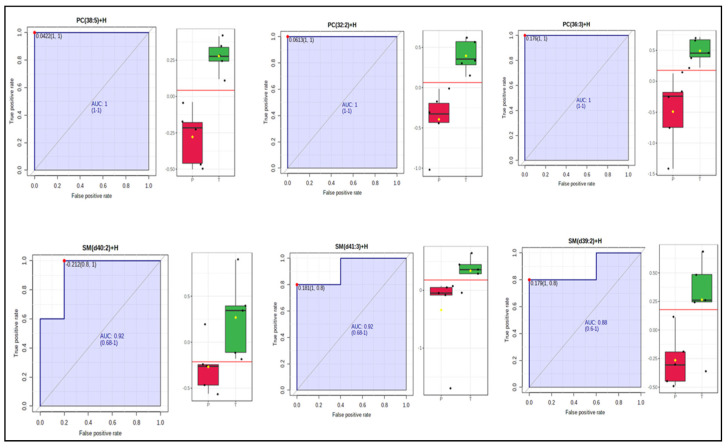
Univariate ROC analysis of selected significantly changed lipids (PC 38:5, PC 32:2, PC 36:3, SM d40:2, SM d41:3, and SM d39:2) that increase in benfotiamine-treated serum samples (*p* value < 0.05). Red boxess represent placebo samples, green boxes are BFT treated samples and yellow diamond means the median value for log_2_ (fold change).

**Figure 8 ijms-22-13188-f008:**
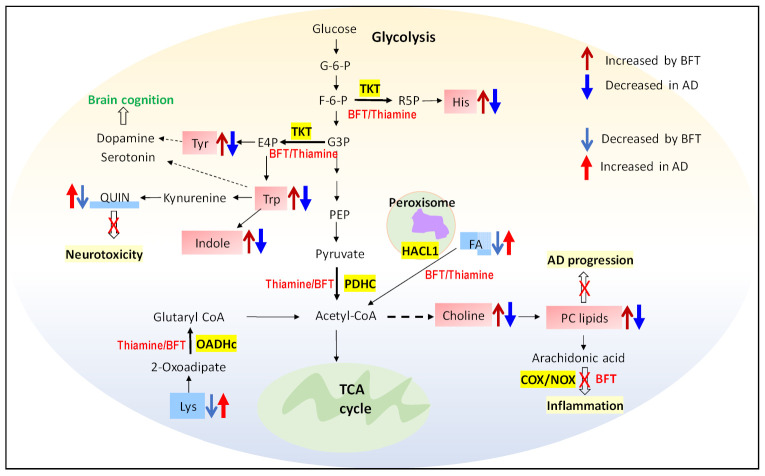
A schematic diagram represents proposed mechanistic pathways in response to benfotiamine treatment. Increased metabolites (light red boxes) and decreased metabolites (light blue box) are shown due to benfotiamine treatment in mild AD individuals. Highlighted yellow boxes represent the enzymes that are thiamine dependent and are responsible for the overserved changes of metabolites and lipids in benfotiamine treatment, which results in reversing the development of AD pathology. FA: fatty acid, Quin: quinoline, PC: phosphatidylcholine.

**Table 1 ijms-22-13188-t001:** List of the 25 metabolites that changed significantly by the largest amount in AD serum after BFT treatment and comparison with those reported in literature for AD progression.

Primary ID	VIP Score	FC of T/P	log_2_(FC) of T/P	*p* Value	BFT Treatment	Literature FC for AD/C [28,29,30,31]
** *Thiamine* **	** *2.974* **	** *101.07* **	** *6.66* **	** *0.0110* **	↑	0.18↓
***4-[(Z)-2-Aminovinyl]phenol* ***	** *1.273* **	** *2.16* **	** *1.11* **	** *0.0306* **	↑	NA
***3-Buten-1-amine* ***	** *1.121* **	** *1.77* **	** *0.82* **	** *0.0036* **	↑	NA
** *Tyrosine* **	** *1.329* **	** *1.66* **	** *0.73* **	** *0.0496* **	↑	0.86↓
** *Betaine* ** *****	** *1.128* **	** *1.641* **	** *0.71* **	** *0.0154* **	↑	NA
** *Indole* **	** *1.244* **	** *1.59* **	** *0.67* **	** *0.0182* **	↑	0.86↓
** *Tryptophan* **	** *1.518* **	** *1.55* **	** *0.63* **	** *0.0323* **	↑	0.75↓
Choline	1.466	1.77	0.83	0.0632	↑	NA
Histidine	1.174	1.29	0.37	0.0626	↑	0.83↓
Platelet-activating factor	1.035	1.10	0.14	0.4183	↑	NA
1-Palmitoyl-2-hydroxy-sn-glycero-3-PE	1.235	0.69	−0.54	0.3572	↓	NA
** *O-Ureido-D-serine* **	** *1.222* **	** *0.78* **	** *−0.36* **	** *0.0019* **	↓	NA
Carboxymethyl-L-lysine	1.653	0.73	−0.45	0.5331	↓	1.37↑
9-HpODE	1.027	0.71	−0.49	0.1830	↓	NA
10-Undecenoic acid	1.274	0.50	−1.01	0.1024	↓	NA
Palmitic acid	1.150	0.50	−1.01	0.1033	↓	0.65↓
N-Undecanoylglycine	1.054	0.50	−1.01	0.1024	↓	NA
** *Androsterone sulfate* **	*1.219*	** *0.48* **	** *−1.06* **	** *0.0343* **	↓	1.93↑
Uric acid	1.117	0.43	−1.22	0.1340	↓	1.20↑
9(S)-HpOTrE	1.249	0.41	−1.28	0.1733	↓	NA
Linoleic acid	1.351	0.37	−1.43	0.0817	↓	1.28↑
5α-Dihydrotestosterone glucuronide	1.461	0.37	−1.45	0.1190	↓	NA
** *16-Hydroxyhexadecanoic acid* **	*1.220*	** *0.25* **	** *−2.01* **	** *0.0241* **	↓	NA
2-Isopropylmalic acid	1.605	0.15	−2.73	0.2055	↓	NA
** *Lysine* **	** *1.777* **	** *0.08* **	** *−3.56* **	** *0.0272* **	↓	1.28↑

* Represents they are either phytochemical or drug metabolites. VIP represents variable importance in projection, VIP scores were obtained from OPLS-DA model, and the cutoff VIP score is >1. The changed metabolites with additional *p*-values < 0.05, are in bold and italic. FC represents fold change, log_2_(FC) represents log_2_(fold change), BFT-benfotiamine, T-Treated samples, P-Placebo samples, AD-Alzheimer’s disease samples, C-Control samples or healthy ones, NA-Not available. Red arrow-increased level, and blue arrow-decreased level.

**Table 2 ijms-22-13188-t002:** List of 22 lipid ions with significant difference between placebo and benfotiamine-treated groups and comparison with those reported in the literature for AD progression.

Primary ID	VIP	FC of T/P	log_2_(FC) of T/P	*p* Value	BFT Treatment	Literature FC for AD/C [32,33]
**PC(32:2) + H**	**2.08683**	**3.65**	**1.87**	**0.0176**	↑	NA
**PC(32:1) + H**	**1.72063**	**2.78**	**1.47**	**0.0079**	↑	--↓
**PC(40:6) + H**	**1.69282**	**2.67**	**1.42**	**0.0003**	↑	--↓
**PC(38:5) + H**	**1.71809**	**2.55**	**1.35**	**0.0032**	↑	NA
**PC(40:4) + H**	**1.61852**	**2.48**	**1.31**	**0.0055**	↑	--↓
**PC(34:4) + H**	**1.8971**	**3.00**	**1.58**	**0.0035**	↑	NA
**PC(36:3) + H**	**2.13179**	**3.58**	**1.84**	**0.0023**	↑	0.90↓
**PC(36:5) + H**	**1.07928**	**2.27**	**1.18**	**0.0414**	↑	0.59↓
**PC(34:3) + H**	**1.60953**	**2.20**	**1.14**	**0.0003**	↑	--↓
TG(48:3) + NH4	1.4194	1.78	0.83	0.0808	↑	--↓
PC(38:6) + H	1.51857	1.75	0.80	0.3515	↑	0.78↓
**SM(d39:2) + H**	**1.19304**	**1.73**	**0.79**	**0.0064**	↑	--↑
**SM(d41:1) + H**	**1.05188**	**1.49**	**0.57**	**0.0113**	↑	--↓
TG(48:1) + NH4	1.52317	1.39	0.48	0.2737	↑	--↓
**SM(d40:2) + H**	**1.09519**	**1.30**	**0.38**	**0.0271**	↑	NA
TG(58:2) + NH4	1.6588	2.83	1.50	0.405	↑	NA
**TG(53:6) + NH4**	**1.16343**	**0.77**	**−0.37**	**0.098**	↓	>4.0↑
**TG(54:0) + NH4**	**1.06078**	**0.84**	**−0.26**	**0.0454**	↓	NA
TG(58:7) + NH4	1.24209	0.68	−0.55	0.0736	↓	>4.0↑
TG(54:6) + NH4	1.31772	0.67	−0.57	0.0907	↓	NA
**TG(60:10) + NH4**	**1.12928**	**0.61**	**−0.71**	**0.0517**	↓	>4.0↑
TG(54:6) + H	1.04225	0.73	−0.438	0.00958	↓	--↑

VIP represents variable importance in projection, VIP scores were obtained from the OPLS-DA model and the cutoff VIP score is >1. The changed metabolites with additional *p* values < 0.05, are in bold. FC represents fold change, log_2_(FC) represents log2(fold change), BFT-benfotiamine, T-Treated samples, P-Placebo samples, AD-Alzheimer’s disease samples, C-Control samples or healthy ones. NA-Not available. Red arrow-increased level and blue arrow-decreased level. --↓ and --↑ indicate changes, no defined values being reported by literature.

## Data Availability

The data presented in this study are available on request from the corresponding author. The data are not publicly available yet and we are working on data repository.

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
