# Peer review of "Serum Metabolomic and Lipidomic Profiling Reveals Novel Biomarkers of Efficacy for Benfotiamine in Alzheimer’s Disease"

_ijms, 2021, doi:10.3390/ijms222413188_

Round 1

Reviewer 1 Report

The work of Bhawal and colleagues presented for review concerns the metabolomic and lipidomic analysis of the serum of Alzheimer's patients undergoing benfotiamine therapy. The issue raised is very relevant nowadays. We still do not have adequate treatment against neurodegenerative changes, and life expectancy continues to increase. Therefore, the topic taken is the most up-to-date. The research was conducted on a tiny group (5 placebo and 5 treated), which is not yet the final result. The authors themselves write about further research and validation of the results in a larger group.

Benfotiamine is currently in clinical trials, and the authors do not mention it in the manuscript. It is worth pointing out. The layout of the work, figures and tables are prepared carefully and adequately discussed. The language of the work is scientific and does not require corrections.

My comments on the manuscript mainly relate to the Materials and Methods section.

The work is carried out on serum samples, 10 in total (5 attempts per group). The samples were taken at the time, I assume 0 and after 12 months as described. However, there is no information about the processing time. Were the serum samples taken at the beginning analyzed right away, were they frozen and waited 12 months for the final samples? How were they secured? It is vital information for the analysis of metabolites. Has the effect of long-term storage (if starting samples waited 12 months for -80-degree analysis) on matrix changes been tested? The authors do not address this problem at all.

The authors described the methods used, each time referring to the work mentioned in the references. Of course, there is now a tendency to shorten and not add self-plagiarism to manuscripts, but some basic information should be included. In the case of tests using mass spectrometry, it is essential to specify the MS instrument and the setup, i.e. what chromatographic system supported the given spectrometer. Of course, you can guess it or find the mentioned work, but it would be easier for people who are not experts in mass speculation to check it in the manuscript you read. Similarly, in the case of metabolomics and lipid urinary studies, the columns used are given. Still, only their type, without manufacturer and dimensions - if you want to recreate or compare your data to those in the manuscript, it is good to know the exact materials used.

It is now a good practice to put the resulting output in public repositories. There is no mention of it in the publication, and I recommend that you deposit your data and provide an ID / DOI number allowing the readers to view the data.

The authors analyzed the data against their library for C18 and HILIC workflow and mentioned they found 315 unique metabolites not listed in any public database. I am asking you to develop this idea of what repositories and how many authors in general have more IDs in their library than public databases. Is the created database planned to be made available to a broader group of scientists? Maybe it is worth building a network for sharing raw data and databases to increase the chances of finding markers and validating them? Internal standards were added during the processing of samples, but there is no information, for example, about the recovery percentage during processing.

Reviewer 2 Report

Bhawal R, et al., have presented an interesting study on serum metabolomics and lipidomics profiling of Alzheimer's patients treated with Benfotiamine. There are some comments concerning the work:

  1. The authors enrolled mild dementia patients with AD in this study. However, the ADAS-Cog score of the individuals included in this study is not mentioned. Did the treated individuals in this study have an improvement after treatment?
  2. The sample size is the study is low (5 treated and 5 placebo). A previous study from this group (PMID: 33074237) had a higher sample size. Is it feasible to report similar results for higher sample size?
  3.  The authors identify some phosphatidylcholine (PC) and sphingomyelins (SM) species having significant difference between placebo and treatment group. Were the peaks for these species matched with standard libraries or the in-house spectral libraries?
  4. The authors have not mentioned the IRB number for their study.
  5. At some places that authors have mislabeled green and red dots/. For example in Figure 4, placebo is red and treatment is green, but in the text, it is written differently.

Author Response

Bhawal R, et al., have presented an interesting study on serum metabolomics and lipidomics profiling of Alzheimer's patients treated with Benfotiamine. There are some comments concerning the work:

Response: We thank the reviewer for providing us valuable suggestion to enhance our manuscript focused on serum metabolomics and lipidomics profiling of Alzheimer's patients treated with Benfotiamine.

  1. The authors enrolled mild dementia patients with AD in this study. However, the ADAS-Cog score of the individuals included in this study is not mentioned. Did the treated individuals in this study have an improvement after treatment?

Response:

The initial ADAS-Cog score of the whole population of 70 patients was 15.34 (SD 6.36) (see Gibson et al., PMID: 33074237). Thus, the patients were mildly cognitively impaired or mild AD. The increase in ADAS-Cog was 43% lower in the Benfotiamine group than in the placebo group, indicating less cognitive decline, and this effect was nearly statistically significant (p = 0.125). However, the ADAS-Cog was only one of six measures that we used to measure cognition.  For example, the CDR was significantly changed.

To relate the changes in cognition in our small sample size of 5 serum samples from the placebo and 5 from the Benfotiamine group would create statistical artifacts.

The results show the effects of treating patients with Benfotiamine for one year.  All the results are completely novel. This small study will inform subsequent choice for biomarkers.  The current plan is for the study to now be done in 406 patients. Performing the measures we established in the current study could be done in those patients to see if changes correlate to improvement.

Although we cannot draw conclusion whether the treated individuals in this study have an improvement after treatment or not, we have shown exciting biomarkers for planned studies and the long-term consequences of a one-year treatment with Benfotiamine.    

  1. The sample size is the study is low (5 treated and 5 placebo). A previous study from this group (PMID: 33074237) had a higher sample size. Is it feasible to report similar results for higher sample size?

Response: We agree that the sample size of the study is small and have emphasized that in the manuscript. Unfortunately, the availability of serum sample volume limited this study to only 5 and 5 from the original 34 (Benfotiamine group) and 36 (placebo group) conducted in the paper (PMID: 33074237). Hence, we emphasized the fact that this is an exploratory study to provide a foundation to make a convincing case for expanding the measures in an expanded study in future studies.

  1. The authors identify some phosphatidylcholine (PC) and sphingomyelins (SM) species having significant difference between placebo and treatment group. Were the peaks for these species matched with standard libraries or the in-house spectral libraries?

Response: Phosphatidylcholine (PC) and sphingomyelins (SM) species were found to be significantly different between placebo and treatment groups with very low p value. The untargeted lipidomics data files were searched by LipidSearch software against an online MS/MS Lipid library.  The identity of lipids is based on the MS and MS/MS fragmentation spectra matching to the annotated lipid species by LipidSearch software. Therefore, we are confident in their identification and relative quantitation between the two groups.

The in-house spectral libraries were used only for untargeted metabolomics analysis in this study.

Change Made: we replaced a sentence of “Acquired MS/MS data was processed using LipidSearch™ software version 4.2 (Thermo Scientific) using the same workflow.” in Section 4.4 with the following sentence in the revised version: “Acquired MS/MS data was processed using LipidSearch™ software version 4.2 (Thermo Scientific) against an online MS/MS Lipid library.

  1. The authors have not mentioned the IRB number for their study.

Response: We do have an IRB approved protocol number for this study and the published pilot clinal trial study, which is BRC-451. Please see the attached approval letter from The Burke Rehabilitation Hospital.

Change Made: we added the IRB approved protocol number (BRC-451) to the 4.1 section in a sentence: “…… exploratory measures were approved by the IRBs with the protocol number: BRC-451.

  1. At some places that authors have mislabeled green and red dots/. For example in Figure 4, placebo is red and treatment is green, but in the text, it is written differently.

Change Made:  In the text of Section 2.3, we corrected treatment with green dots and placebo with red dots to match with what shown in Figure 4.

Reviewer 3 Report

Manuscript deals with metabolomics and lipidomics profiling of human serum in patients affected by AD that underwent plpacebo/treatment with BFT for 12 months.

This is an explorative study, as correctly highlighted by authors, as 5 subject are not sufficient to provide more comprehnesive results. This study, however, provides new insight in AD progression and the effect of BFT treatment.

I don't have much to ask to authors, just some minor corrections.

1) In the abstract, authors stated the 417 lipids were quantified. However, this is an untargeted approach, a semi quantitative approach. I'd correct quantified with semi-quantified.

2) it is not necessary to repeat vitamin B1 after thiamin, as it is clearly stated in the irst line of introduction

3) same with (LC) in the second line of paragraph 2.1

4) please check definition of OPLS-DA. To me it is Partial Least Square and not Linear (page 4)

5)add space between p and value (p-value)

Author Response

Manuscript deals with metabolomics and lipidomics profiling of human serum in patients affected by AD that underwent plpacebo/treatment with BFT for 12 months.

This is an explorative study, as correctly highlighted by authors, as 5 subject are not sufficient to provide more comprehnesive results. This study, however, provides new insight in AD progression and the effect of BFT treatment.

I don't have much to ask to authors, just some minor corrections.

Response: The authors are grateful for the reviewer’s recognition of the importance of our exploratory study.

  • In the abstract, authors stated the 417 lipids were quantified. However, this is an untargeted approach, a semi quantitative approach. I'd correct quantified with semi-quantified.

Response: Thanks for the reviewer’s point. We did untargeted approach for lipidomics. It is basically relative quantitation between the two groups.

Change Made: We have changed the “quantified” to “relatively quantified” in the Abstract section.

  • it is not necessary to repeat vitamin B1 after thiamin, as it is clearly stated in the irst line of introduction

Change Made:  We deleted the “(vitamin B1)” in the revised manuscript.

  • same with (LC) in the second line of paragraph 2.1

Change Made:  We deleted the “(LC)” as suggested.

  • please check definition of OPLS-DA. To me it is Partial Least Square and not Linear (page 4)

Change Made: “linear” was changed to “least”.

       5) add space between p and value (p-value)

Change Made: in both Table 1 and Table 2, we added a space between p and “value”.